# Reconfigurable artificial microswimmers with internal feedback

L. Alvarez [1✉], M. A. Fernandez-Rodriguez [1,2], A. Alegria [3], S. Arrese-Igor [3], K. Zhao[1], M. Kröger [4] & Lucio Isa [1✉]

Self-propelling microparticles are often proposed as synthetic models for biological microswimmers, yet they lack the internally regulated adaptation of their biological counterparts. Conversely, adaptation can be encoded in larger-scale soft-robotic devices but remains elusive to transfer to the colloidal scale. Here, we create responsive microswimmers, powered by electro-hydrodynamic flows, which can adapt their motility via internal reconfiguration. Using sequential capillary assembly, we fabricate deterministic colloidal clusters comprising soft thermo-responsive microgels and light-absorbing particles. Light absorption induces preferential local heating and triggers the volume phase transition of the microgels, leading to an adaptation of the clusters' motility, which is orthogonal to their propulsion scheme. We rationalize this response via the coupling between self-propulsion and variations of particle shape and dielectric properties upon heating. Harnessing such coupling allows for strategies to achieve local dynamical control with simple illumination patterns, revealing exciting opportunities for developing tactic active materials.

[1] Laboratory for Soft Materials and Interfaces, Department of Materials, ETH Zurich, Zurich, Switzerland. [2] Biocolloid and Fluid Physics Group, Applied Physics Department, Faculty of Sciences, University of Granada, Granada, Spain. [3] Centro de Física de Materiales (CSIC-UPV/EHU), Materials Physics Center, San Sebastián, Spain. [4] Polymer Physics, Department of Materials, ETH Zurich, Zurich, Switzerland. ✉email: laura.alvarez-frances@mat.ethz.ch; lucio.isa@mat.ethz.ch

The ubiquity and success of motile bacteria are strongly coupled to their ability to autonomously adapt to different environments as they can reconfigure their shape, metabolism, and motility via internal feedback mechanisms[1,2]. Realizing artificial microswimmers with similar adaptation capabilities and autonomous behavior might substantially impact technologies ranging from optimal transport to sensing and microrobotics[3]. Focusing on adaptation, existing approaches at the colloidal scale mostly rely on external feedback, either to regulate motility via the spatiotemporal modulation of the propulsion velocity and direction[4–8] or to induce shape changes via the same magnetic or electric fields[9–11], which are also driving the particles. On the contrary, endowing artificial microswimmers with an internal feedback mechanism, which regulates motility in response to stimuli that are decoupled from the source of propulsion, remains an elusive task.

A promising route to achieve this goal is to exploit the coupling between particle shape and motility. Efficient switching between different propulsion states can, for instance, be reached by the spontaneous aggregation of symmetry-breaking active clusters of varying geometry[12–15], albeit this process does not have the desired deterministic control. Conversely, designing colloidal clusters with fixed shapes and compositions offers fine control on motility[16–18] but lacks adaptation. Although progress on reconfigurable robots at the sub-millimeter scale has been made[19–23], downscaling these concepts to the colloidal level demands alternative fabrication and design. Shape-shifting colloidal clusters reconfiguring along a predefined pathway in response to local stimuli[24] would combine both characteristics, with high potential toward the vision of realizing adaptive artificial microswimmers. Here, we present an approach to fabricate reconfigurable microswimmers relying on a simple combination of standard "hard" particles with soft responsive colloids.

## Results

**Fabrication of reconfigurable colloidal clusters**. To create our reconfigurable microswimmers, we fabricate geometrically and compositionally asymmetric colloidal clusters containing both polystyrene (PS) microparticles (2 µm Ø, fluorescent Ex/Em–530/607 nm) and soft thermo-responsive microgels (poly-iso-propylacrylamide-co-methacrylic acid[25], PNIPAM-co-MAA, hydrodynamic diameter 1.4 µm in MilliQ water at 23 °C) using sequential Capillarity-Assisted Particle Assembly (sCAPA)[17,26].

In brief, we assemble both particles onto a polydimethylsiloxane (PDMS) substrate that has traps with a geometry corresponding to the target shape of the final cluster. We first deposit the PS particles and, in a second step, we add the microgels, which fill the free space left in the traps after the first deposition (Fig. 1a). The trap-filling process is enabled by the capillary force exerted by the meniscus of an evaporating droplet of the colloidal suspensions driven over the substrate at a controlled speed. The formed PS-microgel clusters are then thermally sintered to ensure their mechanical integrity upon harvest and transferred to the experimental cell (more details in Materials and Methods). We achieve the controlled formation of cluster populations with different geometries on different regions of the same template, e.g., dumbbells and L-shapes, defined by the local orientation of the moving meniscus relative to the trap orientation. The largest population is composed of dumbbells (see Fig. 1b and Supplementary Fig. 1).

The incorporation of the PNIPAM-co-MAA microgels endows the dumbbells with reversible temperature response. These microgels exhibit a volume phase transition (VPT) in water at a temperature $T \simeq 32$ °C[25]. Upon crossing the VPTT, the microgels reduce their volume by ~80% (Supplementary Fig. 2) and

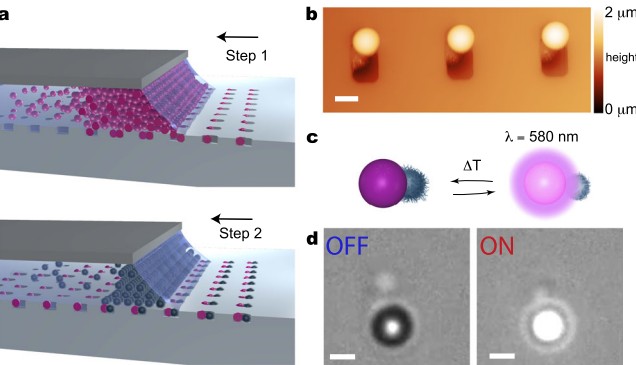

**Fig. 1 Fabrication of thermo-responsive reconfigurable dumbbells. a** Scheme of the deposition of polystyrene (PS) particles (step 1) and PNIPAM-co-MAA microgels (step 2) onto a PDMS template patterned with 2 × 4 µm² size traps. The arrow indicates the direction of the deposition. Particles are accumulated at the meniscus of the moving droplet and deposited into the traps via capillary forces. **b** AFM image in the water showing PS-microgel dumbbells in the traps before harvesting. The bright spheres are the PS colloids and the darker ones are the swollen microgels. **c, d** Schematic representation **c** and optical micrographs **d** of the light-driven reconfiguration of a dumbbell. Fluorescent illumination with a power density $\rho_{FL} = 54$ mW(mm)$^{-2}$ locally increases the temperature above the VPTT of the microgel, causing it to deswell and correspondingly induce a variation of the dumbbell's geometry and dielectric properties. The transition is reversible upon removing the incident light. Scale bars: 2 µm.

drastically change their dielectric properties, essentially switching from a swollen, spherical polyelectrolyte to a more compact, particle-like object[25,27–31]. Over the same $T$-range, the size and dielectric properties of the PS particles remain practically unchanged. However, the PS particles act as the vehicle to induce the reconfiguration of the microgels. Illuminating the dye-loaded PS particles using light with a wavelength of 450–490 nm, where the dye partially absorbs, causes preferential local heating owing to non-radiative energy dissipation (Supplementary Fig. 3). By controlling the fluorescence power density ($\rho_{FL}$) within a thermalized environment (starting from a temperature of 21.3 ± 1.1 °C or of 24.1 ± 0.8 °C) the temperature locally increases to the phase transition point at 32 °C, triggering the reconfiguration of the microgels attached to the PS particles (Fig. 1c, d and Supplementary Movie 1). In particular, for illumination power densities $\rho_{FL} \leq 54$ mW(mm)$^{-2}$ preferential local heating is obtained up to a $T = 35.0 ± 1.5$ °C (Supplementary Fig. 3f). The local $T$ vs. $\rho_{FL}$ is calibrated measuring the enhancement of particle diffusivity. Under the same illumination conditions, only a minor enhancement of diffusivity is measured for non-fluorescent particles (Supplementary Fig. 3e) and correspondingly only a small global temperature increase is measured (Supplementary Fig. 4b). Finally, no discernible photobleaching is seen during the time of the experiments, ensuring a constant local temperature (Supplementary Fig. 3c).

The integrated PS-microgel clusters thus contain the essential elements of a reconfigurable two-state system: the PS particles convert an external light intensity signal into heat, causing the transition of the microgels from a swollen to a collapsed state, which can be reversed by reducing the illumination. When the reconfiguration is coupled to self-propulsion, the result is a reconfigurable microswimmer with internal feedback triggered by the sensing of a light stimulus orthogonal to its propulsion scheme.

**Adaptive propulsion via electrohydrodynamic flows**. In this work, such coupling is reached by driving self-propulsion via

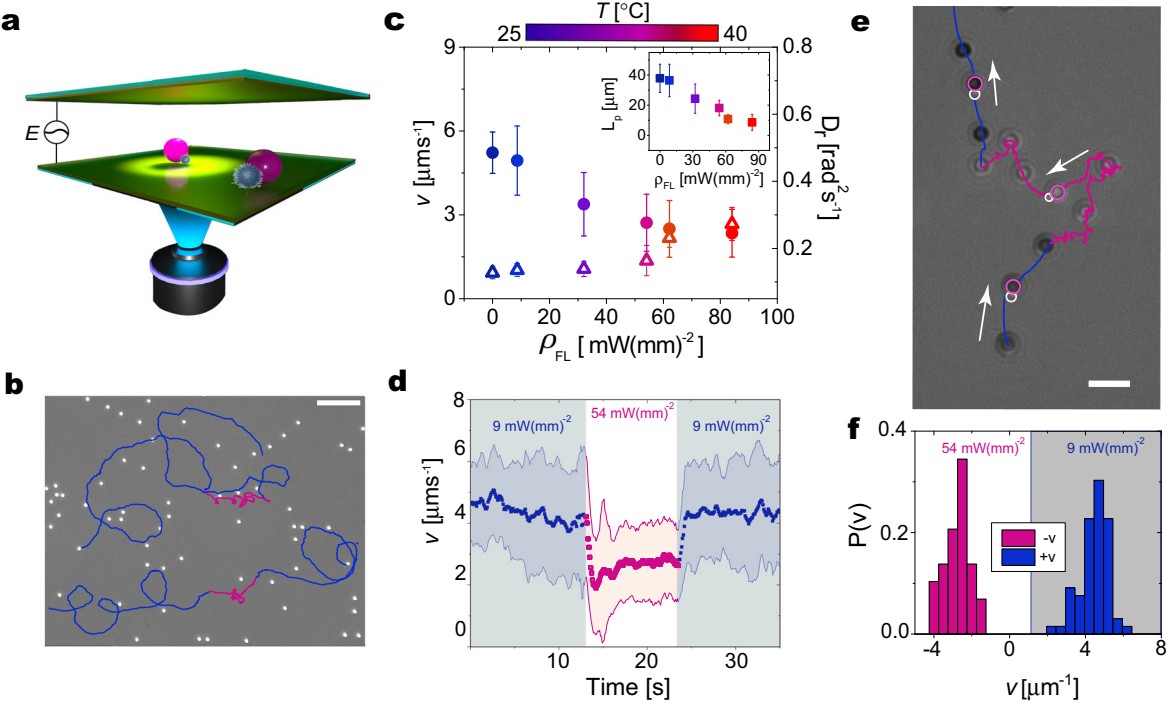

**Fig. 2 Experimental realization of adaptive active dumbbells. a** Schematic representation of the experimental cell, illustrating the transverse AC electric field $E$ and the local illumination with fluorescent light, respectively generating motion and causing particle reconfiguration. **b** Examples of particle trajectories switching $\rho_{FL}$ between 9 (blue trajectory) and 54 mW(mm)$^{-2}$ (magenta trajectory). At $\rho_{FL} = 54$ mW(mm)$^{-2}$, the PS particle heats the microgel above its volume phase transition temperature (VPTT), causing a motility change. **c** $v$ (solid circles) and $D_r$ (open triangles) as a function of illumination power density $\rho_{FL}$. The inset represents the persistence length ($L_p$) of the particles' trajectories as a function of $\rho_{FL}$. The color scale represents the corresponding temperatures. **d** Particle velocity as a function of time for two levels of illumination $\rho_{FL} = 9$ (shaded areas) and 54 (white area) mW (mm)$^{-2}$. Scale bar represents 20 μm. **e** Particle trajectory (×63 magnification) showing that at $\rho_{FL} = 9$ mW(mm)$^{-2}$ (blue) the particle swims toward the PS lobe ($+v$), and at $\rho_{FL} = 54$ mW(mm)$^{-2}$ (magenta) it changes direction and swims with the microgel in front ($-v$). The pink and white circles indicate the position of the PS particle and microgel, respectively. **f** Histogram of particle velocity (45 particles) at low $\rho_{FL}$ (9 mW(mm)$^{-2}$—gray area) and high $\rho_{FL}$ (54 mW(mm)$^{-2}$—white area). Error bars in all cases indicate the standard deviation of the data.

electrohydrodynamic flows (EHDFs)[32]. These flows result from the interaction between the induced dipole of a particle near an electrode and the induced charges in the electric double layer of the substrate in the presence of a transverse AC electric field. They are in particular owing to the emergence of a tangential component of the electric field caused by the distortion of the distribution of accumulated charges by the presence of the particle. These flows are radially symmetric around a homogeneous spherical object, but their symmetry can be broken by breaking the symmetry of the particle shape and/or dielectric properties, e.g., as for the case of a dumbbell composed of two lobes made of different materials. Asymmetric EHDFs lead to net propulsion of the particle relative to the electrode. The direction and the magnitude of the propulsion depend on the contrast in size and dielectric properties of the different lobes[14,33].

In our experiments, we confine an aqueous suspension of the reconfigurable clusters between two conductive transparent surfaces separated by a spacer of thickness $2H$. Upon applying an AC voltage with peak-to-peak amplitude $V_{pp} = 4–5$ V at a frequency $f = 1$ kHz, the clusters self-propel with a swimming velocity $v \propto (V_{pp}/2H)^2$, leading to velocities that vary between 3 and 6 μm s$^{-1}$ without any fluorescence illumination, i.e., at room temperature. The application of the AC electric field with amplitudes $V_{pp} < 6$ V does not lead to global Joule heating (Supplementary Fig. 4b) and temperature is solely controlled by illumination. We track the motion of the particles by means of an inverted microscope (×40 magnification objective at 10 fps) mixing transmission and epifluorescence illumination (Fig. 2a),

where the modulation of the latter source encodes the input signal for particle reconfiguration. More than two-thirds of the clusters are either PS-microgel dumbbells or triangular clusters with one PS particle and two microgels (Supplementary Fig. 1), which show very similar dynamics (Supplementary Fig. 5). We focus our attention on the dumbbells and later present additional features that emerge for L-shaped particles.

The example trajectories depicted in Fig. 2b show that the motility of self-propelling dumbbells adapts to the level of fluorescence illumination. In particular, we see that upon switching from a low ($\rho_{FL} = 9$ mW(mm)$^{-2}$, blue trajectory sections) to a high ($\rho_{FL} = 54$ mW(mm)$^{-2}$, magenta trajectory sections) fluorescence power density, the trajectories show marked qualitative differences, with motion becoming strongly less directed in the magenta sections. We obtain a quantitative assessment of this adaptation by independently measuring $v$ and the rotational diffusivity $D_r$ as a function of $\rho_{FL}$ (Fig. 2c, details in the Methods Section). We observe that $D_r$ increases owing to (i) particle size changes following the microgel collapse, and (ii) heat-induced local viscosity changes (Supplementary Fig. 6). Conversely, the velocity exhibits a marked reduction up to $\rho_{FL} = 54$ mW(mm)$^{-2}$, after which it remains practically constant. Combining these two effects, the persistence length of the trajectories, defined as $L_p = v/D_r$, experiences a threefold decrease for an input illumination that induces the reconfiguration of the microgels collapse (inset to Fig. 1c). A systematic quantification carried out over 400 particles shows that the velocity changes are fully reversible (Supplementary Fig. 7, Supplementary Movies 2 and 3) and that the adaptation to illumination, and hence

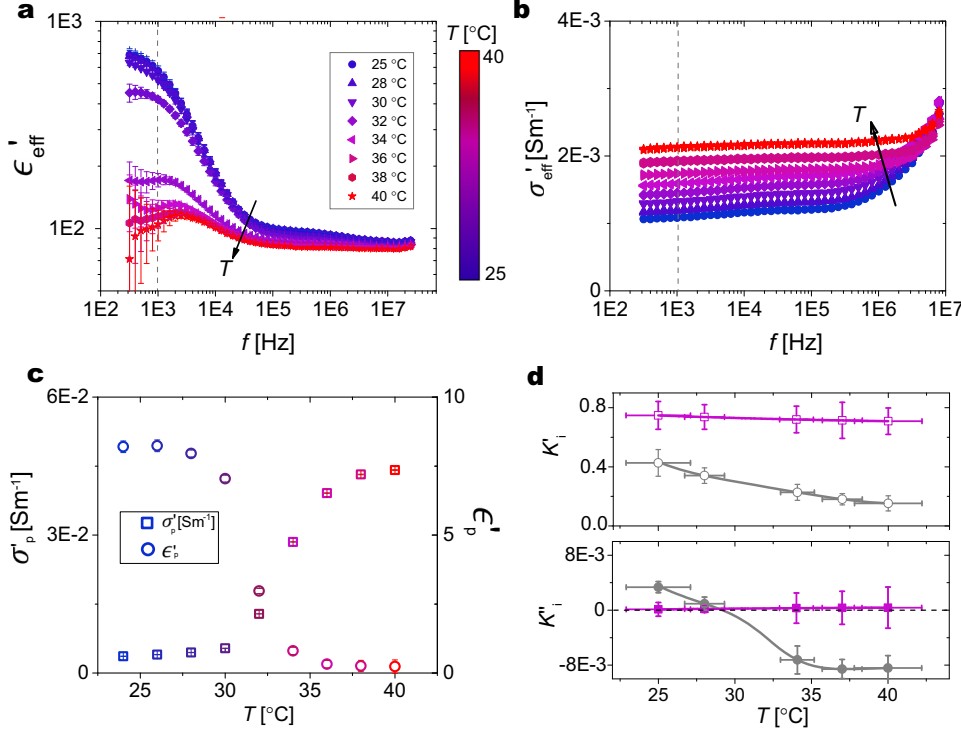

**Fig. 3 Dielectric spectroscopy measurements. a** $\epsilon'_{\text{eff}}$ and **b** $\sigma'_{\text{eff}}$ as a function of frequency $f$ for an aqueous suspension of microgels (1 wt %) measured for $T$ between 22 and 40 °C. The data are corrected to remove electrode polarization. **c** Permitivitty ($\epsilon'_p$, open circles) and conductivity ($\sigma'_p$, open squares) vs. $T$ at $f = 1$ kHz. **d** Real ($K'$ top, open symbols) and imaginary part ($K''$ bottom, solid symbols) of the Claussius–Mossotti factor for each particle $i$, i.e., microgels (gray) and PS particles (pink). The values are obtained for each particle from Eq. (2), using the corresponding values of $\sigma'_p$ and $\epsilon'_p$. The error bars indicate the standard deviation of the data.

temperature, variations has a characteristic response time of a few seconds (Fig. 1d). This finite response time may be related to the timescale of the reconfiguration of microgels adsorbed on the PS particle surface (Supplementary Movie 1). The same experiments performed with dumbbells comprising a non-fluorescent PS particle and a microgel, show no change of the trajectories upon modification of the fluorescence illumination, confirming that reconfiguration is mediated by light absorption (Supplementary Movie 4, Supplementary Fig. 8).

**Temperature-dependent dielectric properties and motion reversal**. A closer inspection of the time dependence of the motility changes reveals a richer dynamical response. At higher magnifications (×63 objective at 10 fps), both the PS and the microgel lobe can be distinguished during the self-propulsion of the dumbbells (Fig. 2e). At low $\rho_{\text{FL}}$, we observe that all particles swim with the PS lobe in front ($+v$). At higher $\rho_{\text{FL}} = 54$ mW (mm)$^{-2}$ we instead observe that the particles invert their direction of motion and start swimming at a different speed with the microgel lobe in front ($-v$) (Fig. 2f, Supplementary Movie 5).

The direction reversal and the speed change can be rationalized by investigating how the microgel's collapse affects the local EHDFs around a PS-microgel dumbbell. The fluid velocity as a function of distance $r$ away from the surface of each spherical particle "i" comprising the dumbbell is given by

$$U_i = \frac{C}{\eta} \frac{K'_i + K''_i \bar{\omega}}{1 + \bar{\omega}^2} \frac{3(r/R_i)}{2\left[1 + (r/R_i)^2\right]^{5/2}}, \quad (1)$$

where $\eta$ is the solvent viscosity, $R_i$ is the particle radius, $K'_i$ and $K''_i$ are the real and imaginary part of the particle's Claussius–Mossotti factor $K^*_i$, respectively, and $C = \beta\epsilon_m\epsilon_0 H(V_{\text{pp}}/2H)^2$[33]. Here, $\beta$ is a

constant prefactor used as a single fitting parameter to obtain the experimental velocities, $\epsilon_m$ is the solvent relative permittivity, $\epsilon_0$ is the vacuum permittivity, and $\bar{\omega} = \omega H/\kappa D$ is a normalized angular frequency ($\omega = 2\pi f$, with $f$ the frequency of the applied AC field), with $\kappa$ being the inverse Debye length and $D$ the ion diffusivity in the solvent. By including the temperature dependence of the particle properties, i.e., $R_i$ and $K_i$, as well as the temperature dependence of the solvent viscosity, we can predict the EHDFs around each particle as a function of illumination conditions.

In particular, the Claussius–Mossotti factor $K^*$ represents the complex polarizability of the particles relative to the surrounding medium and thus determines the sign and the magnitude of the EHDFs in Eq. (1). This factor is defined as

$$K^* = \frac{\epsilon^*_p - \epsilon^*_m}{\epsilon^*_p + 2\epsilon^*_m}, \quad (2)$$

where $\epsilon^*_p$ and $\epsilon^*_m$ are the complex permittivities of the particle and the medium, respectively. $\epsilon^*_p$ can be written as $\epsilon^*_p = \epsilon_0(\epsilon'_p - j\epsilon''_p) = \epsilon_0\epsilon'_p - j\sigma'_p/\omega$, where $\epsilon'_p$ is the real part of the relative particle permittivity and $\sigma'_p$ is the real part of the particle conductivity, respectively, and $j = \sqrt{-1}$. For charged colloids in electrolyte solutions, the total particle conductivity can be typically expressed as $\sigma'_p = \sigma_b + 2K_s/R$, where $\sigma_b$ is the bulk conductivity of the particle and $K_s$ is the surface conductance, defined as the sum of the Stern layer conductance and the conductance of the diffuse layer. Previous work has shown that direct knowledge of zeta potential and Stern layer conductivity can be used to predict the magnitude and sign of the EHDFs surrounding charged particles, where bulk effects can be neglected[34]. The values of $\epsilon'_p$ and $\sigma'_p$ for the PS particles are

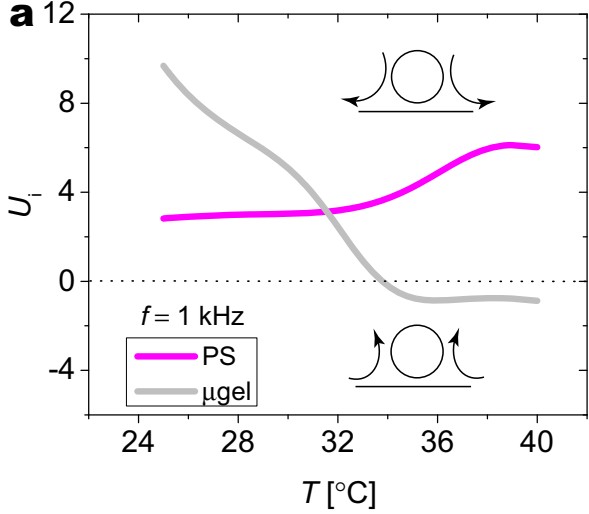

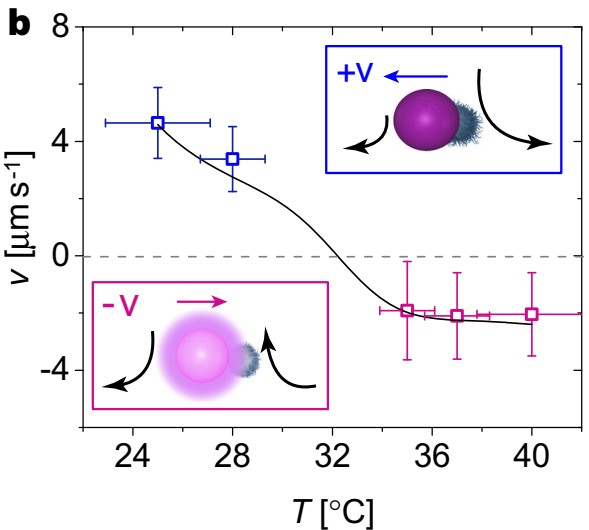

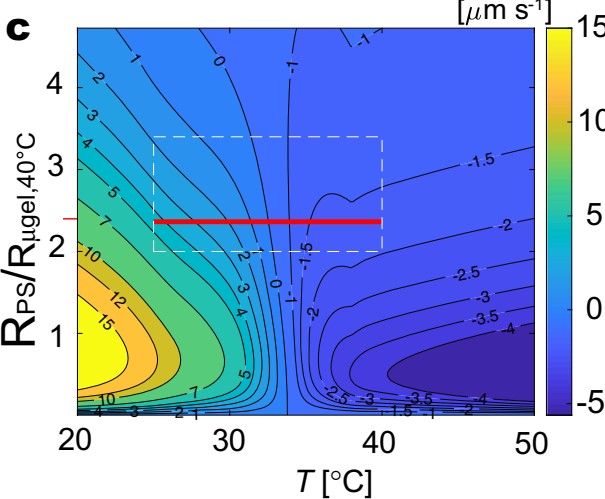

**Fig. 4 Feedback between propulsion mechanism and particle reconfiguration. a** EHDF velocities $U_i$ for each particle calculated from Eq. (1) with the fitting parameters $\beta_1 = 0.23$ (PS) and $\beta_2 = 4.4$ (microgel). The schematics indicate the direction of the EHDFs, being either positive (repulsive) or negative (attractive). **b** Experimental values (symbols) and theoretical prediction (solid line) of dumbbell velocity as a function of temperature. The inset schemes indicate the EHD flows and final propulsion direction. y-error bars are the standard deviation of the velocity calculated for 45 dumbbells at each temperature. x-error bars correspond to the uncertainty in experimentally determining $T$. **c** Theoretical prediction of dumbbell velocity $v$ as a function of $T$ and size ratio between the PS sphere and the collapsed microgel (at 40 °C). The plot is obtained from Eq. (1) using the experimental temperature-dependent system properties (particle size and dielectric properties, solvent viscosity) as inputs and a single-temperature independent prefactor. The dashed red box highlights the experimentally accessible range of $T$ and size ratios (the range of size ratios is estimated from the error bars in Fig. 4b). The red solid horizontal line corresponds to the black solid curve in Fig. 4b.

However, the situation is more complex for the charged microgels. In particular, upon crossing the VPTT, the charge distribution within the particles, and thus the different contributions to their polarizability, strongly change. We expect to assist in a transition from a swollen state where charges are distributed throughout the microgel's hydrated volume to a state where charges are mainly located in the periphery of the particle upon the microgel collapse[27]. In both cases, the presence of a so-called "fuzzy surface" of loosely cross-linked polymer chains also renders the location of the Stern layer and of the counterion cloud experimentally poorly defined[36]. Finally, the different swelling state of the microgel below and above the VPTT leads to a swelling- and crosslinking-density-profile-dependent electrophoretic mobility, as described by Ohshima's theory[37], such that simple zeta potential measurements cannot be used to extract the surface conductivity without detailed knowledge of the microgel conformation and charge distribution[38].

To circumvent these uncertainties on the relative roles of the different contributions to the microgels' polarizability, we used dielectric spectroscopy to extract the $\epsilon'_p$ and $\sigma'_p$ of a single microgel from the direct measurements of the effective $\epsilon'_{eff}$ and $\sigma'_{eff}$ of a microgel suspension as a function of frequency and temperature. These measurements give us direct access to the Clausius–Mosotti factor, which, as previously introduced in Eq. (1), is the quantity required to evaluate the direction and magnitude of the EHDF velocity as a function of frequency and temperature. Figure 3a, b shows the $\epsilon'_{eff}(f)$ and $\sigma'_{eff}(f)$ for a microgel suspension measured in the frequency range $10^2 - 10^7$ Hz for temperatures between 22 °C and 40 °C, crossing the VPTT of 32 °C. The single-particle dielectric properties follow the same trend after a renormalization by the suspension's volume fraction (see Supplementary Note 1), and, for simplicity, we refer to those in the following discussion. In particular, at low frequency (including the 1 kHz used in our experiments), the permittivity signal is dominated by the polarization of the electrodes, but after appropriately subtracting this contribution (Supplementary Fig. 10), $\epsilon'_p(f)$ clearly shows a temperature dependence (Fig. 3a). By increasing temperature, we observe an overall reduction of the permittivity down to values of and consequently a dramatic decrease in the particle permittivity $\epsilon'_p$, with a particularly marked drop happening at ~32 °C. At higher frequencies, not relevant in our experiments, different relaxations of the permittivity are seen in the swollen and collapsed state (Supplementary Fig. 10), corresponding to different polarization mechanisms, as described

obtained following these calculations based on literature inputs, as described in Supplementary Notes 1 and 2. In particular, $\epsilon'_p$ is essentially constant in the experimental temperature and frequency ranges[35], whereas for $\sigma'_p$ for PS at 1 kHz shows a moderate increase (Supplementary Fig. 11).

in the literature[28,30,31,39]. Correspondingly, $\sigma'_p$ shows a gradual increase with increasing temperature (Fig. 3b).

The temperature-dependent values of $\epsilon'_p$ and $\sigma'_p$ at a frequency of 1 kHz are shown in Fig. 3c. The growth of conductivity is associated with an increased charge density following the particle's size reduction at higher temperatures[31]. In contrast, the rapid drop of permittivity is the result of a sudden decrease of the particle size and of the corresponding drop of mobility of the charges distributed throughout the volume of the microgel, which also contributes at these low frequencies[39], leading to an increased role of surface effects.

With these inputs, we can now compute the temperature dependence of $K'$ and $K''$ for both particle types, as shown in Fig. 3d. Although we note that $K'$ only shows a weak temperature dependence for both particles, with the values of the PS being systematically greater than the ones of the microgels, the situation changes for $K''$. Here, the values for the PS particles are almost constant and always positive, but $K''$ for the microgels changes signs from positive to negative upon crossing the VPTT.

Using the experimental values of $K'$, together with temperature-dependent values of $R_{\mu gel}$ and $\eta$ (see Supplementary Information), we can now predict $U_i(T)$ for each lobe of the dumbbell, evaluated at a position $r$ corresponding to the center of the other lobe, based on Eq. (1)[33]. The calculation shows that $U_{\mu gel}$ decreases and inverts its direction as $T$ grows and crosses the VPTT, as a consequence of the size change and the sign reversal of $K''$. Correspondingly, $U_{PS}$ exhibits a slight increase, owing to a reduction of the fluid viscosity in the vicinity of the

heated particle and to the change of $r$ at which it is evaluated (Fig. 4a). Combining the $T$-dependent velocities of the EHDFs for each particle, the dumbbell velocity $v(T)$ can, in first approximation, be obtained as a linear combination of the two values of $U_i(T)$ as

$$v(T) = \frac{U_{\mu gel}(T)R_{PS} + U_{PS}(T)R_{\mu gel}(T)}{R_{\mu gel}(T) + R_{PS}} \qquad (3)$$

As we see in Fig. 4b, this simple prediction matches the experimental data very well, capturing both the inversion of propulsion direction and the change in magnitude across the VPTT. At low-light intensities, the overall behavior is dominated by the repulsive flows around the microgel surface, causing the dumbbell to propel with the PS lobe in front. When the microgel shrinks at higher light intensities, the EHDF close to its surface changes signs and is strongly reduced. This EHDF variation causes an inversion of the propulsion direction, where the now stronger repulsive flows generated by the PS lobe propel the dumbbell with the microgel in front. The reversal of the propulsion is also further supported by direct visualization of the EHDFs around each lobe of the dumbbells using fluorescent tracers (750 nm), and by examining the assembly of PS particles and microgels while applying the actuating AC field and changing the global temperature conditions[14,34] (Supplementary Figs. 12 and 13, and Supplementary Movies 8–9).

The velocity prediction can be extended to different size ratios between the dumbbell's lobes, as shown in Fig. 4c, with the solid

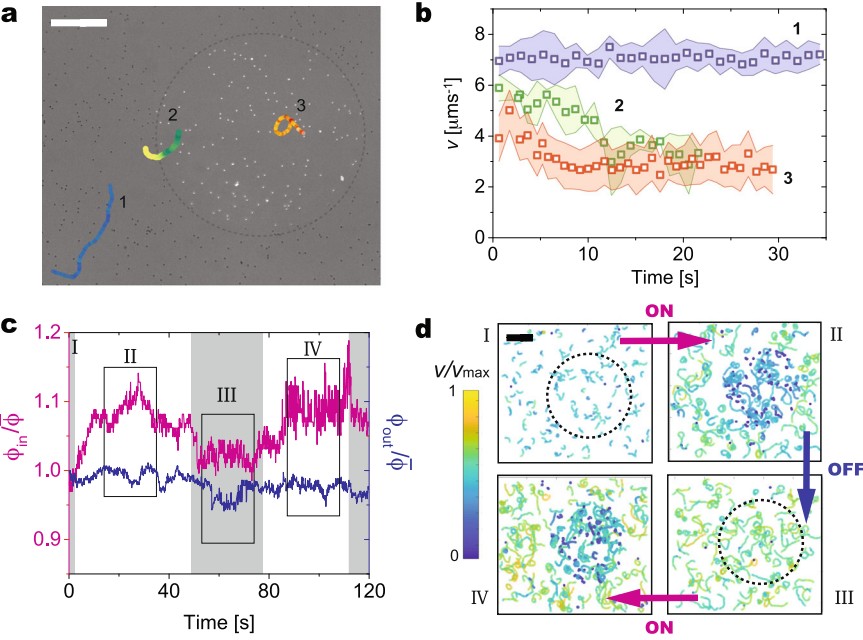

**Fig. 5 Adapting swimming to light patterns. a** Combined transmission and epifluorescent micrograph of self-propelling dumbbells where the fluorescent light ($\rho_{FL} = 0.2$ mW(mm)$^{-2}$) is confined within the dashed circle by partly closing the microscope diaphragm. Characteristic trajectories of active particles remaining outside (light blue to dark blue) or inside (red to yellow) the illuminated region, or crossing from one to the other (yellow to green). The trajectories are color-coded based on the instantaneous velocity. Upon entering the illuminated region, the dumbbell clearly slows down. Scale bar: 150 μm. **b** Particle velocities (absolute values) as a function of time for the three representative particles depicted in (**a**), using the same colors. Time $t = 0$ s corresponds to the fluorescent light being turned on. The shaded bands represent the error bars corresponding to the standard deviations of the instantaneous particle velocity over 0.4 s (four frames). **c** Particle number density inside $\phi_{in}$ (red–left axis) and outside the illuminated region $\phi_{out}$ (blue–right axis) versus time during on (white) and off (gray) cycles of the fluorescent light. The number density is normalized by the initial number density, $\bar{\phi}$, of a uniform particle distribution before turning on the fluorescent light. The data are produced by cumulating three independent experiments. **d** Particle trajectories during the on-off cycles corresponding to the boxed regions in (**c**) color-coded by instantaneous velocity (averaged over four frames) for 50 frames (I) and 200 frames (II, III, IV). During the on cycles, the particles within the illuminated region significantly slow down and the persistence of their trajectories drops. During the off cycles, they return to their original swimming behavior. The particles outside the illuminated region are not affected. Scale bars represent 40 μm.

red line corresponding to the $T$-range for our experiments shown in Fig. 4b. This parametric representation offers promising guidelines for the design of dumbbells with a tailored velocity modulation and shows how the response is very sensitive to small variations close to the VPTT. Among the different parameters, it emerges that the changes in the dielectric properties of the microgels across their VPTT play the dominant role in regulating the dumbbell's dynamical response. Calculations for different frequencies are reported in the Supplementary Information (Supplementary Fig. 14).

**Position- and shape-dependent dynamics.** The presence of light-driven reconfiguration enables us to modulate the particle swimming behavior using simple light patterns. By partly closing the diaphragm of the fluorescence illumination, we can create circular regions with controlled values of $\rho_{FL}$ (Fig. 5a) and track the motion of active dumbbells inside or outside these regions, or crossing between the two. We find that particles remaining outside the illuminated region (blue data in Fig. 5b) consistently show a constant higher velocity than the ones inside the circular light pattern (red data), where $t = 0$ s corresponds to the time when the fluorescence is turned on. The particles inside the illuminated region show that the adaptation of the propulsion velocity has a characteristic time scale of a few seconds (see also Fig. 1d), analogously experienced by a particle that enters the illuminated region and progressively adapts to a different propulsion speed (green data).

The position-dependent swimming imparted by the light patterns affects the collective behavior of ensembles of reconfigurable dumbbells. In particular, particle accumulation inside the illuminated regions can be elicited as a consequence of a progressive slowing down of particles entering the illuminated regions[40,41]. Cyclic on-and-off illumination causes a reversible increase of the particle number density inside the illuminated circle during the "on" times, which returns to the same value of the number density outside the circle when the fluorescent light is switched off (Fig. 5c). The density changes for $\phi_{in}$ (inside) are clearly correlated to a decrease of particle propulsion speed and persistence of the trajectories within the illuminated region, as seen by the particle trajectories (Fig. 5d). The propulsion of the particles outside the illuminated circle remains unaffected.

Finally, the coupling between internal reconfiguration and propulsion offers even more opportunities if the complexity of the cluster shape is enhanced. Here, we show that sCAPA-fabricated L-shaped clusters comprising two PS spheres on the long arm and a PS-microgel dumbbell on the short arm (Fig. 6a), show curvilinear trajectories with a light-switchable chirality (Supplementary Movies 6 and 7). L-shaped active particles are known to exhibit curvilinear trajectories whose pitch only depends on shape[42,42] and which are confined within a 2D plane defined by the balance between gravity and electrostatic repulsion with the substrate[43]. In our case, by changing the illumination conditions leading to the reconfiguration of the microgel, the L-shaped clusters simultaneously adapt their shape and propulsion velocity, both slowing down and inverting the propulsion direction relative to the orientation of the short arm of the L. As a result, the trajectory displayed in Fig. 6b shows that both the pitch and the chirality of the motion changes from counter-clockwise $\Omega^-$ to clockwise $\Omega^+$, as confirmed by tracking the particle angular position over time (Fig. 6c).

**Discussion**
Our results demonstrate that programmable adaptive behavior can be implemented by exploiting soft responsive colloids as components of artificial microswimmers. In fact, even though the

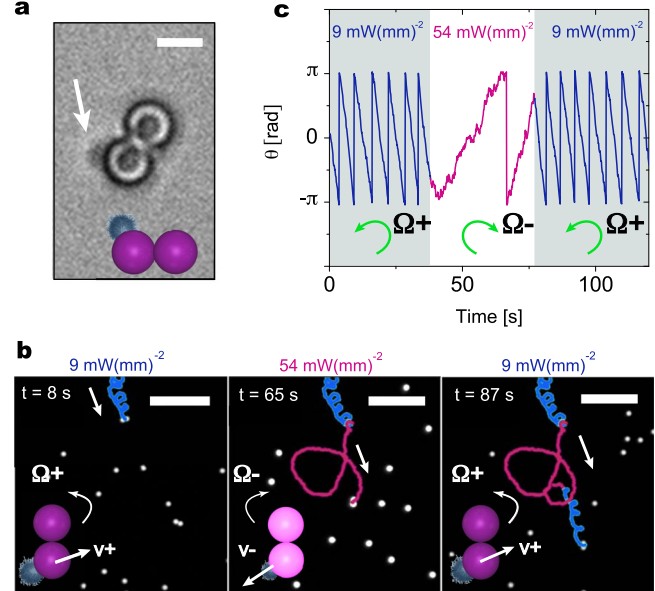

**Fig. 6 Motion chirality control with L-shaped particles. a** Optical microscopy image of an L-shaped reconfigurable cluster comprising two PS particles and one microgel in MilliQ water at room temperature. The arrow indicates the microgel position **b** Trajectories of an L-shaped particle at two different levels of fluorescence light power density ($\rho_{FL} = 9$ mW(mm)$^{-2}$–blue; $\rho_{FL} = 54$ mW(mm)$^{-2}$–magenta). The schematics in the insets show the propulsion direction relative to the PS-microgel combination as a function of $\rho_{FL}$ and the corresponding angular velocity $\Omega$. Upon changing illumination density, the helical trajectory changes chirality, due to propulsion reversal, and pitch, due to shape reconfiguration. Scale bars: 5 µm. **c** Orientation angle as a function of time for a self-propelling L-shaped particle at different light illumination levels. Upon tuning $\rho_{FL}$ up, the rotation changes direction and slows down. The change is fully reversible upon reducing the fluorescent light.

propulsion is driven by externally applied AC electric fields and the reconfiguration is triggered by illumination, the adaptation of motility is internal. In particular, the reconfiguration is not externally guided, but follows a pathway that is encoded in the microswimmer during its synthesis. By controlling the relative particle positions during sequential capillary assembly and by selecting microgels with given properties (e.g., size, swelling ratio, and dielectric properties), the active clusters will spontaneously respond to temperature changes in a programmed fashion. This programmability constitutes an important step toward developing truly autonomous microswimmers, even if autonomy is not yet achieved. In fact, by exploiting conceptually similar adaptation schemes, one can envision self-regulated systems where propulsion is internally generated, e.g., by chemical reactions, and reconfiguration follows spontaneously generated environmental cues, such as dynamically evolving temperature and pH changes associated with the propulsion reaction. Before approaching these more complex realizations, robust feedback schemes are required, where the orthogonality between propulsion and reconfiguration control offers many options. Moreover, through the design and incorporation of soft, responsive elements in deterministic positions, one can endow the particles with internal degrees of freedom, which can be tailored to engineer both single-particle and collective response[44]. In particular, the ability to probe the environment and correspondingly adapt the dynamical behavior under a variety of external cues resonates with the complex dynamical response of biological microswimmers, such as chemotactic[45], light-responsive[46,47], or rheotactic bacteria[48].

Our approach to introducing responsive colloids in micro-swimmer fabrication processes presents exciting opportunities since it offers the option to combine a vast palette of materials from established colloidal synthesis routes into programmable architectures and devices. Future developments of this strategy are closely connected to progress in materials and microfabrication, to equip active particles with multiple responses and functionalities, downsizing the potential of soft robotics to the colloidal scale.

In conclusion, we present an experimental realization of reconfigurable active colloids with internal feedback, coupling variations of shape and material properties to propulsion. In particular, we connect the propulsion of active colloidal clusters to their temperature-dependent size and dielectric properties, mediated by the presence of soft, thermo-responsive microgels, which we extensively characterize.

## Methods

**sCAPA and harvesting of reconfigurable colloidal clusters.** Our active thermo-responsive clusters are formed by a combination of 2 μm PS (Microparticles Gmbh) and PNIPAM-co-MAA microgel particles. The microgels have a hydrodynamic diameter $2R_{\mu gel} = 1.4$ μm at 25 °C, as measured in MilliQ water by dynamic light scattering (Malvern Zetasizer), and have been synthesized by surfactant-free emulsion polymerization[25]. The size response of the microgels was measured by dynamic light scattering (Malvern Zetasizer) from 23 to 40 °C. The clusters were fabricated by sCAPA[17]. In brief, a 45 μL droplet of a 0.1% v/v PS colloidal solution with 0.05 mM SDS (Sigma-Aldrich) and 0.005 wt% Triton X-30 (Sigma-Aldrich) was confined between a template and a flat PDMS piece and dragged at a speed of 3 μm/s and a temperature of 25 °C. The template contains an array of micro-sized rectangular traps of $4 \times 2$ μm and 1 μm-depth over an area of 2 cm². After the droplet passes, traps are filled with one or two PS particles, in the central or lateral region of the template, respectively. The process is then repeated with an aqueous dispersion of PNIPAM-co-MAA microgels in 0.001 wt% of Triton X-30, which are deposited in close contact with the PS particles. The local orientation of the droplet meniscus relative to the traps defines the number and position of deposited microgels as described in the main manuscript. The PS-microgel assemblies are sintered in the oven for 5 minutes at 70 °C. Microgels deposited in the traps but not in contact with PS particles remain disconnected. Finally, the clusters are harvested by freezing a droplet of 5 μL MilliQ water on the traps, and lifting it from the template, diluted in 3 μL of MilliQ water; 7.4 μL of the thawed particle solution are transferred to the sample cell for the experiments, without further treatment.

**Experimental setup.** The active colloidal clusters are imaged in a customized sample cell comprising two transparent electrodes, separated by an adhesive spacer with a 9 mm-circular opening and 0.12 mm height (Grace Bio-Labs SecureSeal, USA). The transparent electrodes were fabricated using 22 mm × 22 mm glasses (85–115 μm-thick, Menzel Gläser, Germany) coated via e-beam metal evaporation with 3 nm Cr and 10 nm Au (Evatec BAK501 LL, Switzerland), and plasma-enhanced chemical vapor deposition with 10 nm of $SiO_2$ (STS Multiplex CVD, UK) to minimize particles sticking at the surface of the conductive glass slide. The electrodes are connected to a function generator (National Instruments Agilent 3352X, USA) that applies the AC electric field, with a fixed frequency of 1 kHz and $V_{pp}$ between 5 and 6 V (32-42 V/mm). Imaging was carried out in an inverted microscope (Axio Observer Z1) using ×20 and ×40 objectives (Zeiss). Image sequences (540 × 650 pixel², 16 bits) were taken with an sCMOS camera (Andor Zyla) at a frame rate of 10 fps and exposure time of 10 ms. The epifluorescence illumination source (North 89) was used to excite the PS particles using a band-pass filter $\lambda = 450$–490 nm for imaging and local heating. Bright-field, transmission illumination was used in combination with epifluorescence to identify both components of the active clusters. Movies at high magnification (×63) were recorded with an inverted Eclipse Ti2-E (Nikon) microscope at 10 fps with 10 ms exposure time.

**Dielectric Spectroscopy measurements.** The dielectric properties of the PNIPAM-co-MAA microgels were measured with a Novocontrol high-resolution dielectric analyzer (Alpha-A). The measurements were performed on a cell with a 6.45 mm gap distance between the electrodes enclosed by a Teflon cylinder, which was filled with the microgel suspension at 1 wt%. The permittivity ($\epsilon'_{eff}$, $\epsilon'_m$), and conductivity ($\sigma'_{eff}$, $\sigma'_m$) of the microgel suspension and the media, respectively, were determined over a wide frequency range ($10^2$–$10^7$ Hz) for temperatures between 25 °C and 40 °C, in steps of 2 °C. The procedure to correct electrode polarization is reported in Supplementary Note 1.

## Data availability

The data that support the findings of this study are available from the corresponding authors upon reasonable request. Source data are provided with this paper.

## Code availability

The code used in this study is available from the corresponding authors upon reasonable request.

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

## Acknowledgements
The authors thank Peter Schurtenberger and Heiko Wolf for insightful discussions, Walter Richtering for providing the microgels, and Philippe Nicollier for assisting with the substrate fabrication. L.I. and L.A. acknowledge financial support from the Swiss National Science Foundation (SNSF) Grant PP00P2-172913/1 and the European Soft Matter Infrastructure (EUSMI) proposal number E190900328. M.K. acknowledges SNCF support through grant 200021L-185052.

## Author contributions
Author contributions are defined based on the CRediT (Contributor Roles Taxonomy) and listed alphabetically. Conceptualization: L.A. and L.I.; data curation: A.A., L.A., S.A.I. and K.Z.; formal analysis: L.A., M.K., and K.Z.; funding acquisition: L.I.; investigation: L.A. and M.A.F.-R.; methodology: A.A., L.A., S.A.I., M.A.F.-R., L.I., M.K., and K.Z.; project administration: L.A. and L.I.; software: L.A., M.K., and K.Z.; supervision: L.I.; validation: L.A. and M.A.F.-R.; visualization: A.A., L.A., S.A.I., and L.I; Writing–original draft: L.A. and L.I.; writing–review and editing: A.A., L.A., S.A.I., M.A.F.-R., L.I., and M.K.

## Competing interests
The authors declare no competing interests.
