## [Peer Review File · Nature Communications]

REVIEWER COMMENTS

Reviewer #1 (Remarks to the Author):

The article presents a dumbbell-shaped particles as a self-propelling system which alters its motion via change in properties on one of the lobes in response to external stimuli. The lobes of particles are made of polystyrene and poly-NIPAM, which induces a selective thermoresponsiveness where only one lobe shrinks upon increasing the temperature. Authors used AC electric field to drive the propulsion, and external light to induce a phase transformation in the poly-NIPAM lobe which results in changes in the motion trajectory of the particles. The article presents few interesting findings, but all the effects observed are well known and established. The unique part of the articles in combining these effects, which is non-trivial. My major criticisms come from the overselling of the presented idea and lack of scientific discussion in the article. Therefore, I cannot recommend its publication in present form. My comments/suggestions are as follows:

1. While authors try to make the claim about the autonomy of the particles, I believe it is still a combination of two external stimuli one driving the propulsion and one (sort of) controlling the characteristics of propulsion (which has been previously demonstrated). Therefore, abstract and introduction should be toned down, and authors should refocus these sections on the science and minimize overselling the model system.
2. The authors demonstrate that dumbbell particles invert the direction of swimming from upon increasing the incident light intensity. Why do authors believe that the inversion of the propulsion direction is the result of change in bulk K'' of the microgel and PS? It is known that in AC electric field, the primary contribution to the net polarization (and corresponding fluid flow) originates from the electrical double layer. Since authors state in line 67, that the zeta potential of microgel doubles upon increasing temperature, I would expect the particles to propel with PS facing forward not as reported. Additionally, there is significant change in the volume of counter ionic double layer upon shrinking of microgel, those effects should also be incorporated in the model and corresponding discussion. A clear explanation of these aspects based on the relative surface potential, volume of double layer and polarization of PS and microgel halves is necessary.
3. Was the temperature constant upon the application of the field? And how was it monitored and controlled? The AC-electric fields tend to generate Joule heat, especially with low counter ion concentration (high resistance), which increases the temperature of the dispersion significantly. Given the nature of experiments, it is critical to clarify the effect of field on the dispersion temperature.
4. What is the role of fluid flow at the surface of the experimental chamber? Such flows are known to influence the motion of colloidal particles in AC electric field (as shown by Prof. Ning Wu's work).
5. How do authors know if the motion presented in Fig. 5b is a helix or curvilinear motion confined to a plane? What is the origin of such motion, and what are the effect of gravity, electric field gradients and surface flows on the motion?

Reviewer #2 (Remarks to the Author):

The authors describe the assembly and behavior of artificial microswimmers made from a combination of polystyrene (PS) and polyNIPAM particles and propelled by induced-charge electrophoresis. A shape change and response to the external driving AC field can be triggered by light, through local heating of the PS particle and subsequent collapse of the polyNIPAM particle. This experimental setup is clever and effective. The only minor comment is that I would not consider these particles as "autonomous" microswimmers because they rely on the external field.

The shape-change induced adaption of the motion is conceptually very exciting, although I find the observed change in magnitude and direction of the speed or rotation direction slightly underwhelming. Still, it is a clear effect and the authors demonstrate that this can be used to accumulate particles in a predefined area. The results are sound and well documented, the manuscript is written in a clear manner.

I recommend publication in Nature Communication in its current state.

Reviewer #3 (Remarks to the Author):

The manuscript by Alvarez et al describes a new type of colloidal motors that changes their propulsion speed under AC electric field when the particles are illuminated, as their shape is altered. The work is built upon many previous work where dielectric particles of various shapes can have different response to the electric field, which drive the particle to swim, and where particle shape can shift by some stimuli. The combination of the two seems to work very well in this work, creating a new system that possesses an internal feedback mechanism, resembling examples found in biology. The cool part of the idea is that, although PNIPAM is known to change shape according to temperature, such change here is induced by a nearby fluorescent particle that transfers energy harvested from light. Another point I do not appreciate before reading the paper is that upon changing shapes, the PNIPAM particle's dielectric constant and zeta potential have changed to such an extent that the EHD flow would reverse its direction. This is a key point related to the reversal of the motor's swimming direction. The data presentation and argument are convincing, yet a few of my comments need to be addressed before I would fully support its publication.

1. The energy transfer from the PS to the PNIPAM and raising local temperature is the key concept that makes the system work. I am wondering how this can be verified, and how to rule out the possibility that the whole illuminated area is heated, which does not seem to be uncommon. Some sort of control experiments should be designed: for instance, if PS without dye molecules can be used and under same illumination condition, the propulsion speed of the motor is not changed.

On a related note, in the experiment, is the base temperature of the system controlled, or the experiment is conducted under room temperature? It seems unlikely that, if the experiment is conducted at a low temperature, say 20 degree C, the same fluorescent non-radiative transfer will still be able to heat the PNIPAM above 32 degree C. Will the Au coating on the device contribute to some extent the temperature of the system?

2. As fluorescent particles are used, I am wondering if dye bleaching would cause some inconsistency/problems?

3. ICEO/EHD. The authors introduce the particle propulsion mechanism using ICEO but later focus the discussion on EHD. Although the two describe the same physics, both considering the effect of electric field on ionic charges. However, the two are generally considered different electrokinetic mechanism in the context of micromotors. In papers by Wu. N and Velez. O, these are differently treated, with ICEO describing Janus particle with a gold lobe whereas EHD describes pure dielectric dimers. Their frequency responses also seem different. The authors can consider this point for revision so these terms may be aligned with previous papers.

On a related note, did the authors try a different AC frequency? Will the conclusion still be valid?

4. In supplementary video 3, there seems to be quite a fraction of particles that do not respond/swim. Can the author give an explanation?

5. Conceptually, will simply change the system temperature considered a sort of internal feedback?

Temperature is also an environment parameter that bacteria/particles can feel, then they change shape and response with a different motility.

We report here a detailed response to all questions and comments from the reviewers. The
original comments are in black, our responses in blue. Changes and revisions appear highlighted
in yellow in the manuscript file.

In addition to the revisions to the text, the following figures have been updated:

- • Fig. 2: we have added panels **e** and **f**, and a new color coding for the temperatures at each
ρ_{FL} .
- • Fig. 4: we have taken out panels **a,b** from previous version, and updated panel **c** based on
new calculations with the revised expression Eq. 1.
- • Fig. 5,6: We have updated the color coding of panels **c,d** based on Fig. 2.
- • Fig. S2: the ζ potential has been changed by the electrophoretic mobility μ , and we have
added a color code for the temperatures. We have deleted panel **b** from previous version.
- • Fig. S3: we have changed the order of panels **a,b,c**.
- • Fig. S6-7: we have color coded the temperatures and updated the legend.

And the following figures and the corresponding data have been added:

- • Fig. 3: ϵ'_{eff} and σ'_{eff} as a function of frequency for temperatures between 25-40°C, ϵ'_p
and σ'_p as a function of T and K' and K'' for both particles as a function of T . We have
updated Eq. 1 for the calculation of U_i , to include the full dependence on both the real and
the imaginary part of the Claussius Mossoti Factor K' and K'' .
- • Fig. S4: Control data of the temperature increase of the experimental cell using various
voltages and illumination power density ρ_{FL} conditions.
- • Fig. S8: Control experiments mixing dumbbells with fluorescent and non-fluorescent PS
particles, showing a distinct dynamical behavior as a function of ρ_{FL} .
- • Fig. S9: ϵ'_m and σ'_m for MilliQ water obtained from the dielectric measurements.
- • Fig. S10: Effective permittivity ϵ'_{eff} for the microgel at 25°C and 38°C, as a function of
frequency representing the fitting done for the electrode polarization correction, and its
derivative ϵ'_d .
- • Fig. S11: Estimated surface conductivity of the PS particles calculated from Eq. S5.
- • Fig. S13: Interaction between PS and microgel particles resulting from the EHDFs at high
and low temperature.
- • Fig. S14: Calculated EHDFs velocities for PS particles and microgels and total velocity
of the resulting dumbbell as a function of frequency.

The following Movies have been added: Movie S4, Movie S7, Movie S8 and Movie S9.

**1 Reviewer 1 (Remarks to the Author):**

*The article presents a dumbbell-shaped particles as a self-propelling system which alters*
*its motion via change in properties on one of the lobes in response to external stimuli. The*
*lobes of particles are made of polystyrene and poly-NIPAM, which induces a selective thermore-*
*sponsivenss where only one lobe shrinks upon increasing the temperature. Authors used AC*
*electric field to drive the propulsion, and external light to induce a phase transformation in the*
*poly-NIPAM lobe which results in changes in the motion trajectory of the particles. The article*
*presents few interesting findings, but all the effects observed are well known and established. The*
*unique part of the articles in combining these effects, which is non-trivial. My major criticisms*
*come from the overselling of the presented idea and lack of scientific discussion in the article.*
*Therefore, I cannot recommend its publication in present form. My comments/suggestions are*
*as follows*

We thank the reviewer for the critical reading of the manuscript and for prompting us to ex-
pand on some important aspects of the work. We have now, in particular, expanded the section
relating the temperature-dependent changes of the microgels' dielectric properties to their self-
propulsion. We present new experimental data in the main manuscript and additional estimations
in the Supplementary Information. We have also added a discussion on the notion of autonomy
to place our results into a more precise context.

*1. While authors try to make the claim about the autonomy of the particles, I believe it is still*
*a combination of two external stimuli one driving the propulsion and one (sort of) controlling*
*the characteristics of propulsion (which has been previously demonstrated). Therefore, abstract*
*and introduction should be toned down, and authors should refocus these sections on the science*
*and minimize overselling the model system.*

While we certainly agree with the reviewer that the active motion exhibited by our colloids is
regulated by the combination of two external stimuli, we still maintain that ours is one of the first
experimental realizations of adaptive reconfigurable active particles on the microscale. Import-
antly, the two stimuli are orthogonal. The regulation of the propulsion velocity is not obtained
by a modulation of the stimulus driving it, as in most cases reported in the literature. In fact,
the driving stimulus remains unaltered while the local temperature changes cause the reconfigu-
ration of the particles and thus modify their motion. As mentioned by the referee, this strategy
contributes to the uniqueness of our work.

In response to the reviewer's comment, if by autonomous one considers only closed systems op-
erating without external intervention, then our particles are not complying with this definition.
However, the strategy that we report is in principle applicable to systems where no external in-
tervention is needed, i.e. for chemically-powered particles where reconfiguration is triggered by
spontaneous temperature changes. Nonetheless, the use of external stimuli affords much better
control in this early development phase and we are working towards extending our strategies
to closed systems. We therefore regard our findings as a necessary step to take us closer to the
realization of autonomous microswimmers.

Finally, we remark that the particle reconfiguration is not externally guided during the experi-
ments but it is rather encoded during their fabrication; the particles are designed to reconfigure
along a specific pathway during synthesis and they spontaneously follow this path upon exposure
to the right stimulus. The inclusion of different responsive components during fabrication will
enable the future the design of active particles with more complex spontaneous reconfiguration
pathways.

We have now revised the abstract and introduction and moved a discussion on the above-
mentioned topics to a dedicated section in the conclusions. We hope that our revisions satis-
factorily address the concern of the reviewer.

*2. The authors demonstrate that dumbbell particles invert the direction of swimming from*
*upon increasing the incident light intensity. Why do authors believe that the inversion of the*
*propulsion direction is the result of change in bulk κ of the microgel and PS? It is*
*known that in AC electric field, the primary contribution to the net polarization (and corre-*
*sponding fluid flow) originates from the electrical double layer. Since authors state in line 67,*
*that the zeta potential of microgel doubles upon increasing temperature, I would expect the par-*
*ticles to propel with PS facing forward not as reported. Additionally, there is significant change*
*in the volume of counter ionic double layer upon shrinking of microgel, those effects should*
*also be incorporated in the model and corresponding discussion. A clear explanation of these*
*aspects based on the relative surface potential, volume of double layer and polarization of PS*
*and microgel halves is necessary.*

We thank the reviewer for prompting us to clarify the description and interpretation of our ob-
servations. We have now significantly expanded that section of the manuscript adding new
experimental data and an extensive discussion. As we now explicitly discuss in the manuscript,
the treatment of the different contributions to the polarization of the microgels under different
swelling conditions require particular care and cannot rely on the same assumptions made for
standard charged colloids in electrolyte solutions. In particular, without a detailed knowledge
on the charge distribution across the microgel volume and on its surface, in addition to the lo-
cation of the counter-ion cloud relative to an ill-defined fuzzy surface at the periphery of the
microgels, the precise quantification of the different components of the polarization is elusive.
To avoid making unverifiable assumptions, we measure the total complex permittivity of the
microgels and use these experimental inputs to calculate their complex Clausius-Mosotti factor
as a function of temperature. This, together with the corresponding Clausius-Mosotti factor of
the polystyrene particles, which we now explicitly calculate in the Supplementary Information,
is what is required to estimate the propulsion velocity of our dumbbells under the different illu-
mination conditions. The detailed treatment is found in the manuscript, but the main conclusion
is that the propulsion reversal is to be ascribed to a sudden drop of the microgel's permittivity
across the volume phase transition, as previously reported in the literature. We have also added
new experiments in the Supplementary Information, which confirm that the EHDFs remain re-
pulsive for the PS particles as a function of temperature, while they change sign and become
slightly attractive for the microgel upon collapse (see Figs. S12 and S13).

3. Was the temperature constant upon the application of the field? And how was it mon-
itored and controlled? The AC-electric fields tend to generate Joule heat, especially with low
counter ion concentration (high resistance), which increases the temperature of the dispersion
significantly. Given the nature of experiments, it is critical to clarify the effect of field on the
dispersion temperature.

We have now included additional data on this point in the Supplementary Information (Fig.S4).
The temperature remains constant during the experiments as the sample cell is in contact with
the large heat sink constituted by the microscope operated in a temperature-controlled labora-
tory. In order to support this statement, the global temperature of the dispersion is measured
during the experiment. We now report the measured temperatures collected by directly placing a
thermocouple on the bottom electrode of the experimental cell and recording it as a function of
illumination power density and applied voltage (see Fig. S4). The temperature remains constant
at $24.1 \pm 0.8^\circ\text{C}$ for illumination power densities up to 0.4 mW/mm^2 after which it starts increas-
ing. By fixing the illumination conditions to $\rho_{\text{FL}} = 0.4 \text{ mW/mm}^2$, i.e. sufficient to trigger the
microgel collapse by local temperature increase but not enough to heat the sample globally, we
measure the sample's temperature while applying the AC field within the range of voltages $V_{\text{pp}} =$
$1\text{--}10\text{V}$ ($10\text{--}64 \text{ V/mm}^2$). As the data shows, there is negligible Joule heating within the range
of the voltages applied during the main experiments (up to 5 V), and the dispersion temperature
slightly increases when going above 6 V. However, it stays significantly below the volume phase
transition temperature of the microgels.

4. What is the role of fluid flow at the surface of the experimental chamber? Such flows are
known to influence the motion of colloidal particles in AC electric field (as shown by the work
of Prof. Ning Wu).

As the referee indicates, one of the key elements behind the motion of the colloidal particles is
the proximity of the latter to the bottom electrode. When applying the AC field, mobile ions
accumulate on the surface of the electrode. The resulting electroosmotic flows between the
particle and the bottom electrode, i.e. the electrohydrodynamic flows (EHDFs), arise from the
distortion of such accumulated charges by the local electric field around the particle generated by
its polarization. The magnitude and direction of such flows depends on the dielectric properties
of the particles, given a fixed substrate material and the separation between the particle and the
electrode. We hope that the new extended discussion on the EHDFs in the manuscript clarifies
this issue.

5. How do authors know if the motion presented in Fig. 5b is a helix or curvilinear motion
confined to a plane? What is the origin of such motion, and what are the effect of gravity, electric
field gradients and surface flows on the motion?.

We thank the reviewer for identifying our erroneous nomenclature. At the applied AC field
frequencies and amplitudes, the motion of the L-shapes is confined to a plane and exhibits a
curvilinear/trochoid-like motion.^[1,2] The origin of this motion is to be found in the existence

of a shape-dependent torque (as described in the publications cited above) stemming from a
misalignment between the centre of propulsion and the centre of hydrodynamic drag. Both
gravity and the applied AC field keep the particle aligned flat against the bottom electrode, at a
small separation given by the balance of forces. Previous reports on catalytic swimmers identify
separation values of the order of 100 nm.^[3,2] We have now revised the text to correct the mistake
and clarify the point raised by the reviewer.

**2 Reviewer 2 (Remarks to the Author):**

*The authors describe the assembly and behavior of artificial microswimmers made from*
*a combination of polystyrene (PS) and polyNIPAM particles and propelled by induced-charge*
*electrophoresis. A shape change and response to the external driving AC field can be triggered*
*by light, through local heating of the PS particle and subsequent collapse of the polyNIPAM par-*
*ticle. This experimental setup is clever and effective. The only minor comment is that I would*
*not consider these particles as “autonomous” microswimmers because they rely on the*
*external field. The shape-change induced adaption of the motion is conceptually very exciting,*
*although I find the observed change in magnitude and direction of the speed or rotation direction*
*slightly underwhelming. Still, it is a clear effect and the authors demonstrate that this can be*
*used to accumulate particles in a predefined area. The results are sound and well documented,*
*the manuscript is written in a clear manner. I recommend publication in Nature Communication*
*in its current state.*

We thank Reviewer #2 for their positive comments on our work. The reviewer is correct in
pointing out that the activity of the active particles arises from an external field. As already dis-
cussed in response to a similar comment raised by Reviewer 1, we have now modified the text to
include a more precise discussion on the implications of our results on the notion of autonomous
microswimmers. The second point raised by the reviewer on the moderate change of magni-
tude during motion reversal remains a feature inherent to the microgel’s dielectric properties.
However, our results present exciting opportunities to design composite particles using micro-
gels with different sizes and responses to amplify these effects. For instance, the predictions on
velocity changes reported in Fig. 4c of the revised manuscript indicate that an effective stoppage
of active motion can be achieved by choosing the right size ratios of particles in a dumbbell.
These findings serve thus the purpose to guide the future development of new reconfigurable
swimmers, which we are currently pursuing in our group.

**3 Reviewer 3 (Remarks to the Author):**

*The manuscript by Alvarez et al describes a new type of colloidal motors that changes their*
*propulsion speed under AC electric field when the particles are illuminated, as their shape is*
*altered. The work is built upon many previous work where dielectric particles of various shapes*
*can have different response to the electric field, which drive the particle to swim, and where*

*particle shape can shift by some stimuli. The combination of the two seems to work very well in*
*this work, creating a new system that possesses an internal feedback mechanism, resembling ex-*
*amples found in biology. The cool part of the idea is that, although PNIPAM is known to change*
*shape according to temperature, such change here is induced by a nearby fluorescent particle*
*that transfers energy harvested from light. Another point I do not appreciate before reading*
*the paper is that upon changing shapes, the PNIPAM particle's dielectric constant and zeta*
*potential have changed to such an extent that the EHDflow would reverse its direction. This is*
*a key point related to the reversal of the motor's swimming direction. The data presentation*
*and argument are convincing, yet a few of my comments need to be addressed before I would*
*fully support its publication.*

We thank Reviewer 3 for their positive feedback and constructive comments.

*1. The energy transfer from the PS to the PNIPAM and raising local temperature is the*
*key concept that makes the system work. I am wondering how this can be verified, and how*
*to rule out the possibility that the whole illuminated area is heated, which does not seem to be*
*uncommon. Some sort of control experiments should be designed: for instance, if PS without*
*dye molecules can be used and under same illumination condition, the propulsion speed of the*
*motor is not changed.*

We now present a set of control experiments and direct measurements of the temperature of the
dispersion during fluorescent illumination and applied AC fields. The data reported in Fig. S3
show that in a mixture of fluorescent PS particles and non-fluorescent silica particles, only the
diffusivity of the former colloids changes upon illumination using fluorescent light up to $\rho_{FL} \lesssim$
0.4 mW/mm^2 . This demonstrates that the heating effect is local and limited to the proximity of
the light-absorbing PS particles until global heating of the dispersion takes place. The absence
of a global temperature increase for the above-mentioned range of power densities is confirmed
by the direct measurement of the dispersion temperature using a thermocouple connected to the
bottom electrode, as shown in Fig. S4. Moreover, Fig. S4 also shows that there is negligible
Joule heating upon application of the AC field for the voltages used in the experiments. In direct
response to the reviewer's comment, we have also carried out further control experiments, where
we mixed hybrid microgel-PS dumbbells with one part of the population having a fluorescent
PS sphere and the other part having a non-fluorescent one. As shown in Fig. S8, only the
dumbbells with the fluorescent PS particle change their trajectories and corresponding dynamics
upon fluorescent illumination under a constant AC field. The propulsion velocity of the non-
fluorescent dumbbells stays constant for power densities below the one causing global heating
of the sample (Fig. S8 and Movie S4).

*On a related note, in the experiment, is the base temperature of the system controlled, or*
*the experiment is conducted under room temperature? It seems unlikely that, if the experiment*
*is conducted at a low temperature, say 20 degree C, the same fluorescent non-radiative transfer*
*will still be able to heat the PNIPAM above 32 degree C. Will the Au coating on the device con-*
*tribute to some extent the temperature of the system?*

The base temperature of the experiment is constantly monitored and controlled by: 1) control-
ling the environmental temperature of the whole microscopy set-up, which is inside an enclosed
box at a constant temperature of $23.9 \pm 1^\circ\text{C}$, 2) directly measuring the temperature of the cell by
placing a thermocouple on it, which stays at $24.1 \pm 0.8^\circ\text{C}$ for $\rho_{\text{FL}} \lesssim 0.4 \text{ mW/mm}^2$ as used in our
experiments (Fig. S4).

Finally, the diffusion experiments reported in Fig. S3 (carried out with the Au coating) show that
the diffusivity of non-fluorescent particles remains constant within the illumination boundaries
described above, and we therefore infer that the overall temperature also stays constant under
these circumstances. We remark that we use an excitation wavelength of 450–490 nm, for which
the fluorescent dye presents significant absorption (the absorption spectrum from the manufact-
urer shows half the absorption relative to the maximum at 530 nm), while the Au film does not.
Much stronger heating of the gold film is achieved by irradiating with green light, as exploited
to generate the data shown in Figs. S12 and S13.

*2. As fluorescent particles are used, I am wondering if dye bleaching would cause some in-*
*consistence/problems?*

We have performed control experiments without AC field where we record the emitted fluores-
cence intensity of our polystyrene beads at $\rho_{\text{FL}} = 0.4 \text{ mW/mm}^2$ as a function of time. The data
shows that no significant bleaching is observed up to 500 s (interval between frames of 5 sec-
onds for 500 s). In the experiments such illumination power density is used for up to 2 minutes.
The stability of the dye is also supported by the consistent recovery of initial velocities during
many ON-OFF illumination cycles as shown in Fig. S7. We have now included a discussion
on this point in the manuscript, supported by the additional data included in the Supplementary
Information (Fig. S3a).

*3. ICEO/EHD. The authors introduce the particle propulsion mechanism using ICEO but*
*later focus the discussion on EHD. Although the two describe the same physics, both consid-*
*ering the effect of electric field on ionic charges. However, the two are generally considered*
*different electrokinetic mechanism in the context of micromotors. In papers by Wu, N and Velev,*
*O, these are differently treated, with ICEO describing Janus particle with a gold lobe whereas*
*EHD describes pure dielectric dimers. Their frequency responses also seem different. The au-*
*thors can consider this point for revision so these terms may be aligned with previous papers.*

We thank the reviewer for their comment on the terminology. We agree with their statement and
revised the text accordingly.

*On a related note, did the authors try a different AC frequency? Will the conclusion still be*
*valid?*

The experiments reported this paper have all been carried out at 1kHz and we have focused
on the temperature dependence rather than on the frequency dependence. Initial tests at other

frequencies in the 500-1500 Hz range showed no qualitative differences. However, based on
the dielectric spectroscopy measurements, we can calculate the expected propulsion velocities
at different frequencies using the measured values of the Clausius-Mosotti factor. These calcu-
lations are now reported in Fig. S14. Essentially, the propulsion reversal is maintained but the
overall propulsion speed rapidly decays to zero for frequencies approaching 10^4 Hz. We have
now added a comment in the main text and refer to the data in the Supplementary Information.

*4. In supplementary video 3, there seems to be quite a fraction of particles that do not re-*
*spond/swim. Can the author give an explanation?*

The particles that do not respond or swim might not have a microgel attached (fabrication defect)
and therefore not generate an asymmetric EHD flow, or they may simply stick to the substrate
due to local imperfections of the silica coating.

*5. Conceptually, will simply change the system temperature considered a sort of internal*
*feedback? Temperature is also an environment parameter that bacteria/particles can feel, then*
*they change shape and response with a different motility.*

As we now discuss in the conclusions section of the manuscript, we expect that a conceptually
similar behavior is to be expected in a system where temperature variations are not externally in-
duced, e.g. as we do by illumination, but stem from internal, spontaneous changes which are dy-
namically coupled to self-propulsion. Finally, we remark that we consider temperature changes
as a stimulus, either internally generated or externally induced, but the feedback emerges from
the particle reconfiguration, which we control during particle synthesis, and which proceeds
spontaneously upon the sensing of the temperature stimulus.

**References**

[1] Kümmel, F. *et al.* Circular motion of asymmetric self-propelling particles. *Phys. Rev.*
*Lett.* **110**, 198302 (2013). URL [https://doi.org/10.1103/PhysRevLett.110.](https://doi.org/10.1103/PhysRevLett.110.198302)
[198302](https://doi.org/10.1103/PhysRevLett.110.198302).

[2] ten Hagen, B. *et al.* Gravitaxis of asymmetric self-propelled colloidal particles. *Nat. Com-*
*mun.* **5**, 4829 (2014). URL <https://doi.org/10.1038/ncomms5829>.

[3] Simmchen, J. *et al.* Topographical pathways guide chemical microswimmers. *Nat. Commun.*
**7**, 10598 (2016). URL <https://doi.org/10.1038/ncomms10598>.

REVIEWERS' COMMENTS

Reviewer #1 (Remarks to the Author):

I thank the authors for carefully addressing my concerns and comments. The manuscript is much improved and now provides sufficient discussion on the scientific principles driving the propulsion. I believe the manuscript will generate interesting discussion on the topic and will be of interest to wide audience in active colloids community.

Reviewer #3 (Remarks to the Author):

I have examined the revised manuscript. The authors have addressed my previous comments and concerns in details with new experiments, extended discussions, and supplemented figures and movies etc. I have also looked through other reviewers' comments, which the authors have responded to properly, especially with the added discussion on the mechanism accounting for the reversal of the velocity of the swimmer.

I am satisfied with the current version and therefore recommend its publication.